# Perceived gender equitable norms and previous tuberculosis testing in Malawi: A secondary analysis of a cluster-based prevalence survey

Elizabeth Di Giacomo[1]*, Emily S. Nightingale[1], Peter MacPherson[1,2,3], Helena R. A. Feasey[4], Rebecca N. Soko[2], Vincent K. Phiri[2], Elizabeth L. Corbett[5], Katherine C. Horton[1]

1 Department of Infectious Disease Epidemiology, London School of Hygiene and Tropical Medicine, London, United Kingdom, 2 Malawi Liverpool Wellcome Trust Clinical Research Programme, Blantyre, Malawi, 3 School of Health and Wellbeing, University of Glasgow, Glasgow, United Kingdom, 4 School of Medicine, University of St. Andrews, St. Andrews, United Kingdom, 5 Department of Clinical Research, London School of Hygiene and Tropical Medicine, London United Kingdom

* edigiac3@jh.edu

## Abstract

Substantial evidence demonstrates that men have a higher prevalence of tuberculosis (TB) and decreased use of TB services compared to women. Gender roles and norms contribute to these disparities by influencing social and structural determinants, as well as individual behaviours. In this analysis, we investigated attitudes towards gender equitable norms and TB testing behaviours amongst Malawian men and women participating in a prevalence survey conducted before a community-based TB active case finding trial in Blantyre. Attitudes towards equitable gender norms were captured through a modified version of the Gender Equitable Men Scale (GEMS). Gender inequitable views were prevalent among both men (56.1%) and women (55.8%). The association between a composite GEMS score and TB testing history was modelled using logistic regression, accounting for various sociodemographic covariates (age, sex, wealth quantile, education, and HIV status) (OR = 1.12, 95% CI: 0.88-1.42, p = 0.373). Bivariate analysis demonstrated no notable confounding by any covariates and no strong effect modification. While GEMS score had no association with TB testing history among women, men with higher GEMS scores (less gender-equitable views) were more likely to have been tested for TB across age groups. These findings provide a basis for future investigation into the patterns and motives TB behaviours, particularly in older men. Tailored public health strategies may then be implemented to address this important population.

**Data availability statement:** The data that support the findings of this study are available on Open Science Framework (DOI 10.17605/OSF.IO/GY9JT). Due to ethical restrictions surrounding threats to confidentiality, some data has not been made publicly available.

**Funding:** PM, KCH, and ELC are supported by the UK FCDO ("Leaving no-one behind: transforming gendered pathways to health for TB"). KCH is also supported by the US National Institutes of Health (R-202309-71190). ESN is funded by the Polio Research Committee (WHO; ref: 1432457-1). This research has been partially funded by UK aid from the UK government (to PM, ELC and KCH); however, the views expressed do not necessarily reflect the UK government's official policies. The funders had no role in study design, data collection and analysis, decision to publish, or preparation of the manuscript.

**Competing interests:** The authors have declared that no competing interests exist.

## Introduction

Tuberculosis (TB) remains a persistent global health and development challenge, with progress towards achieving the United Nations Sustainable Development Goal to end the global TB epidemic by 2030 far off track [1]. An estimated 10.8 million people fell ill with TB globally in 2023, with 1.25 million deaths, making it the deadliest infectious disease among adults [2]. From 2005-2019 there was a steady decrease in global TB incidence, however this trend reversed in 2020 and 2021 resulting in large reductions in notifications and associated increases in TB mortality [3]. However, this fluctuation has slowly begun to stabilize [3].

Men are disproportionately affected by TB, accounting for 55% of all people who developed TB in 2023 and 52% of deaths [2]. Prevalence surveys in high burden countries have shown that the underlying burden of undiagnosed TB is over twice as high among men as women, and comparisons of these findings to case notifications indicate that men are less likely than women to access timely diagnosis and treatment [4]. The excess burden of disease and limited access to care among men may be attributed to social and structural determinants that increase men's risk of disease, whilst limiting their ability to access diagnosis and treatment [5]. These range from individual behaviours such as tobacco use and alcohol consumption, to social contact patterns and support networks, to social protections and institutional structures. Underlying many of these determinants are masculine social norms and expectations [4–9]. Consequently, numerous calls to action are being made to address gender discrepancies in TB care, though subsequent action has been limited [10]. Studies are beginning to uncover the specific challenges men face when seeking TB treatment, with recent work exploring the perceptions of masculinity and stigma surrounding TB among men [8,9].

Gender norms are considered important social determinants of health and have been defined as *"a subset of social norms that relate specifically to gender differences. They are informal, deeply entrenched and widely held beliefs about gender roles, power relations, standards or expectations that govern human behaviours and practices in a particular social context and at a particular time"* [11–13]. Research has highlighted how masculine expectations of physical strength, stoicism, and control lead men to avoid admissions of illness and neglect TB symptoms until they become incapacitating, and how men deprioritise their own healthcare needs to fulfil roles as leaders and providers [5,9,14,15]. Evidence associating men's healthcare decisions with masculine norms and expectations suggests gender norms may influence TB testing behaviours, but such an association has not been explored quantitatively [16]. We hypothesized that masculine expectations that affect health behaviours may be linked to attitudes towards gender norms, and that more equitable gender attitudes may be associated with improved access to TB services.

In this secondary analysis of data from a TB prevalence survey conducted as part of a community cluster-randomised trial in Malawi, we investigated the association between attitudes towards equitable gender norms and self-reported TB testing history using a validated tool to measure gender attitudes. We considered relevant covariates, and potential modification of this relationship by age and sex. A

better understanding of the relationship between attitudes towards equitable gender norms and previous TB testing could identify new approaches and methods to reach men, in particular, with TB diagnostic services [17–19]. This could result in more efficient public health strategies to improve diagnosis and treatment of individuals with TB and ultimately reduced transmission in the fight against TB, particularly among men as an underserved population.

## Materials and methods

### Ethics statement

Ethical approval for this secondary analysis of prevalence survey data (reference: 28700) – and for the SCALE Study (reference: 16228) – was obtained through the London School of Hygiene and Tropical Medicine's Research Ethics Committee. In Malawi, ethics approval was obtained from the Malawian College of Medicine and Research Ethics Committee (COMREC, protocol number: P.12/18/2556). All participants provided written (or witnessed thumbprint, if illiterate) informed consent to participate in the parent study.

The data that support the findings of this study are available on Open Science Framework (DOI 10.17605/OSF.IO/GY9JT). Due to ethical restrictions surrounding sensitive participant information, some data has not been made publicly available.

### Study setting and design

We conducted a secondary analysis of data from a community TB prevalence pre-intervention survey in Blantyre, Malawi, implemented from June 2019 to March 2020 before the active case finding cluster-randomised trial "Sustainable Community-wide Active Case Finding for Lung hEalth (SCALE)" (ISRCTN11400592) [17,18]. For the prevalence survey, households were randomly sampled from community neighbourhood clusters, and all adults (≥18 years) were invited to participate in an interviewer-administered questionnaire and TB screening. A 20% random sub-sample of participants were additionally administered an extended questionnaire that included questions on TB testing history, attitudes towards gender equitable norms, healthcare utilization and social contacts mixing (S1 Fig). Inclusion criteria for the original study from which these data were drawn were age of 18 years or older on the day of the interview; able to provide written or witnessed informed consent; intention to remain resident in Blantyre for at least 8 weeks following recruitment; and be willing to have radiological/microbiological screening for TB.

**Measurements and procedures.** For this analysis, the primary outcome was the proportion of participants who self-reported ever testing for TB. The primary outcome measure was participants' response to two pre-intervention survey items on history of TB testing. The first item asked *"have you ever had a chest x-ray (CXR)"*. The second item asked *"have you ever given sputum for a medical test"*. Possible responses to each question were either "yes" or "no".

The primary exposure was individual attitudes towards gender equitable norms, measured using the Gender Equitable Men Scale (GEMS) [19], which provides a validated, standardised framework for quantifying gender attitudes. While the full GEMS consists of 42-items with domains covering violence, sexual relationships, reproductive health and disease prevention, and domestic chores and daily life [20], the SCALE Trial employed a modified 13-item version of the scale, available in both English and Chichewa (see S1 Table). The scale is adaptable, and different versions have been used in a variety of settings [20]. This particular adaptation was informed by previous work in Malawi investigating men's gender attitudes and HIV risk, and a peer-delivered behavioural intervention on contraceptive uptake [21]. As Pierotti's [21] work added new items to the original GEMS scale, we received written permission to use these selected items in our study.

**Covariates.** The primary questionnaire captured data on participant's age and sex, in addition to wealth indicators, education, marital status, employment, and HIV status. The self-assessed wealth index asked participants to position themselves on a six-step scale from poorest to richest, open to their interpretation and frame of reference. This question was taken from the Malawi 2016/2017 National Integrated Household Survey [22]. The self-assessed wealth index was

used as an individual's attitude surrounding their wealth is likely more influential than objective measurements of durable assets in this context.

All participants were asked to report their HIV and antiretroviral therapy (ART) status, and were offered HIV testing. Individuals who self-reported a positive HIV test result, or ART use, or who tested positive were classified as HIV-positive. Other behavioural or clinical risk factors for TB, such as smoking, alcohol use, diabetes, or malnutrition, were not collected as part of the extended questionnaire and therefore could not be explored in this secondary data analysis.

**Analysis.** Analysis was conducted in Stata V.17.0 [23] and R Statistical software (v4.3.3) [24]. We identified latent constructs underlying the 13 GEMS items through exploratory and confirmatory factor analysis (EFA and CFA). We undertook EFA with the principal factor method due to the non-normal distribution of responses to GEMS items [25]. Any items with a uniqueness value >0.70 were excluded. This recommendation is commonly used as a conservative indicator of low shared variance with other items [26–28]. We applied Promax rotation, and variables with high loadings were assessed to identify themes and inform the meaning of underlying latent factors [29]. In assessing the number of factors to retain, we evaluated eigenvalues, scree plots, and performed a parallel analysis. We did not restrict our assessment to eigenvalues alone, as this technique has been cited as prone to both under- and over-factoring [30].

Using Cattell's scree plot, we plotted the magnitude of eigenvalues from a reduced correlation matrix against the number of factors in descending order. We examined the plot for a substantial drop in the magnitude of eigenvalues and considered retaining the factor at and above this [31]. Finally, we conducted a parallel analysis to benchmark the plotted points against predicted means of eigenvalues by 10,000 repeated sets of random data [25]. We considered any points clearly above this line to represent factors that account for more variance than would be expected by chance [32].

Our CFA approach used standardised structural equation modelling (SEM) to construct a model of a set of predicted covariances between our latent variables and then test whether it is plausible when compared to the observed GEMS items [33]. A post-estimation goodness-of-fit of the model was assessed by conducting a likelihood ratio test (LRT). The LRT chi-square results, the root mean squared error of approximation (RMSEA) (<0.08 acceptable), the comparative fit index (CFI) (>0.90 acceptable), and the Tucker-Lewis Index (TLI) (>0.90 acceptable) were assessed [27]. Subsequently, factor scores were predicted using regression methods from the standardized SEM model [32], and merged into one composite score with equal weighting across all items to form the primary exposure variable for the remainder of the analysis. Higher composite scores can be interpreted as more inequitable attitudes towards gender norms, and vice versa. EFA and CFA were conducted on the same dataset due to limited sample size, which may increase the potential for model overfitting.

To investigate the adjusted associations between exposures and outcomes, we first calculated stratum-specific odds ratios of self-reported TB testing history for each covariate with a corresponding 95% confidence interval, and significance level. We stratified GEMS score by TB testing history, and each covariate. We then constructed a logistic regression model of the principal association between the factor composite GEMS score and self-reported TB testing history, including additional co-variates based on *a priori* hypotheses and associations with GEMS items or TB health outcomes in similar settings [4,5,8,34–36]. We incorporated a three-way interaction between the age, sex, and GEMS score to explore potential modification of the GEMS score association by these characteristics. Based on the fitted coefficients from the second model, we predicted the probability and 95% confidence interval for previous TB testing for illustrative values of age, sex, HIV status, and GEMS score.

## Results

Of the 15,897 individuals enrolled in the SCALE Trial, 2,738 participants across 72 clusters completed the extended questionnaire containing the GEMS survey (S1 Fig). This survey captured 2,278 unique households, with 1,720 households containing only a single participant [17]. For this analysis of the study population who responded to the extended survey, 61% of participants were women (1,664) and 39% (1,074) men (Table 1), with a median age of 28 years (range 18–87

**Table 1. Characteristics of participants by tuberculosis testing history.**

| Characteristic | Total (%) (N = 2,738) | Ever Tested for TB* | | OR (95% CI) | Chi-squared p-value |
|---|---|---|---|---|---|
| | | Yes (%) N = 433 (15.81%) | No (%) N = 2,305 (84.19%) | | |
| *DEMOGRAPHIC CHARACTERISTICS* | | | | | |
| *Sex* | | | | | |
| Female | 1664 | 258 (15.50) | 1406 (84.50) | 1.00 (Reference) | ---------- |
| Male | 1074 | 175 (16.29) | 899 (83.71) | 1.06 (0.86-1.31) | 0.581 |
| *Age Group (years)* | | | | | |
| 18-24 | 1028 | 63 (6.13) | 965 (93.87) | 1.00 (Reference) | ----------- |
| 25-34 | 761 | 93 (12.22) | 668 (87.78) | 2.13 (1.52-2.99) | <0.001 |
| 35-44 | 475 | 121 (25.47) | 354 (74.53) | 5.24 (3.72-7.36) | <0.001 |
| 45-54 | 230 | 72 (31.30) | 158 (68.70) | 6.98 (4.69-10.39) | <0.001 |
| ≥55 | 241 | 84 (34.85) | 157 (65.15) | 8.20 (5.53-12.14) | <0.001 |
| *SOCIOECONOMIC CHARACTERISTICS* | | | | | |
| *Self-reported Wealth Index* | | | | | |
| Step 1 (poorest) | 184 | 25 (13.59) | 159 (86.41) | 1.00 (Reference) | ------------ |
| Step 2 | 654 | 82 (12.54) | 572 (87.46) | 0.91 (0.56-1.48) | 0.707 |
| Step 3 | 1216 | 201 (16.53) | 1015 (83.47) | 1.26 (0.80-1.97) | 0.312 |
| Step 4 | 541 | 103 (19.04) | 438 (80.96) | 1.50 (0.93-2.40) | 0.094 |
| Step 5 | 86 | 14 (16.28) | 72 (83.72) | 1.24 (0.61-2.52) | 0.558 |
| Step 6 (richest) | 26 | 3 (10.71) | 25 (89.29) | 0.76 (0.21-2.73) | 0.677 |
| *Education* | | | | | |
| Never attended School or not completed primary | 511 | 85 (16.63) | 426 (83.37) | 1.00 (Reference) | -------- |
| Primary school or Junior certificate | 1226 | 189 (15.42) | 1039 (84.74) | 1.10 (0.82-1.46) | 0.517 |
| Secondary and Higher Education | 1001 | 159 (15.88) | 842 (84.11) | 1.05 (0.78-1.42) | 0.707 |
| *Literacy (Able to read a newspaper or letter in English or Chichewa)* | | | | | |
| Yes | 2527 | 402 (15.91) | 2125 (84.09) | 1.00 (Reference) | 0.642 |
| No | 211 | 31 (14.69) | 180 (85.31) | 0.91 (0.61-1.35) | |
| **Employment** | | | | | |
| Paid Employee (including piece work and domestic work) | 542 | 118 (21.77) | 424 (78.23) | 1.00 (Reference) | ------------ |
| Self-Employed | 690 | 149 (21.59) | 541 (78.41) | 1.01 (0.76-1.34) | 0.303 |
| Unemployed | 1101 | 142 (12.90) | 959 (87.10) | 0.46 (0.34-0.62) | <0.001 |
| Student and other | 405 | 24 (5.90) | 381 (94.1) | 4.41 (2.76-7.32) | <0.001 |
| *HEALTH FACTORS* | | | | | |
| *HIV Testing History* | | | | | |
| No previous testing | 391 | 27 (6.91) | 364 (93.09) | 1.00 (Reference) | ---------- |
| Previous testing | 2347 | 406 (17.30) | 1941 (82.70) | 2.82 (1.88-4.24) | <0.001 |
| *HIV Status* | | | | | |
| Positive | 329 | 124 (37.69) | 205 (62.31) | 1.00 (Reference) | ---------- |
| Negative | 2292 | 295 (12.87) | 1997 (87.13) | 0.24 (0.19-0.81) | <0.001 |
| Unknown | 117 | 14 (11.97) | 103 (88.03) | 1.09 (0.61-2.09) | 0.775 |
| *Presence Of TB Symptoms (any cough, night sweats, weight loss, and/or fever)* | | | | | |
| No | 2305 | 323 (14.01) | 1982 (85.99) | 1.00 (Reference) | ---------- |
| Yes | 433 | 110 (25.40) | 323 (74.60) | 2.09 (1.63-2.68) | <0.001 |

years). 98.8% (N = 2,706) of participants had complete data for ever testing for TB, GEMS items, and all covariates of interest.

## Outcome characteristics

15.8% (433/2,738) of the total population reported ever testing for TB (Table 1). 15.5% of women reported ever testing for TB, while 16.3% of men reported previously testing (Chi-square p = 0.581). The proportion reporting ever testing was highest among individuals ages 35–44 years, with 25.5% having tested for TB in this age group. This age group had 5.24 times the odds of previous TB testing when compared to those in the lowest age group (95% CI: 3.72-7.36, p < 0.001). Testing proportions were lowest among the youngest age group (18–24 years) with only 14.6% reporting having ever tested. Participant characteristics by HIV status are reported in S1 Table.

Amongst those who reported ever testing for TB, 93.8% also reported ever testing for HIV. This was slightly lower among those who did not report ever testing for TB, with 84.2% reporting also ever testing for HIV (Chi-square p < 0.001). 37.7% of HIV-positive participants reported ever testing for TB, while only 12.9% of HIV-negative participants reported having done so (chi-square p < 0.001). 25.4% of those who reported having any TB symptoms (cough, night sweats, unintentional weight loss, or fever) at the time of questionnaire administration also reported ever testing for TB. Interestingly, 14% of those who reported symptoms at the time of the questionnaire did not report previous testing for TB.

**Gender norms.** Composite GEMS score factor and TB testing history are presented in Table 2. Higher GEMS composite scores can be interpreted as having more inequitable views of gender norms. Overall, there was no evidence of difference in GEMS scores distributions between participants who had, and had not, previously tested for TB (p = 0.115). GEMS scores did differ slightly between HIV-positive and HIV-negative participants, with HIV-positive participants exhibiting more inequitable gender attitudes (p = 0.044). The distribution of GEMS responses did not vary substantially

**Table 2. Odds of prior TB testing, by age group, sex and HIV status.**

| HIV Status | Sex | N | Age Group | n (%) | OR for prior TB testing (95% CI) | p-value |
|---|---|---|---|---|---|---|
| _HIV Negative_ | Female | 1364 | _18-24_ | 514 (37.68) | **1.00 (Reference)** | -------- |
| | | | _24-34_ | 434 (31.82) | 1.46 (0.92-2.31) | 0.109 |
| | | | _34-44_ | 226 (16.57) | 3.21 (2.00-5.15) | <0.001 |
| | | | _44-54_ | 91 (6.67) | 4.23 (2.35-7.62) | <0.001 |
| | | | _>55_ | 99 (7.26) | 3.57 (1.98-6.44) | <0.001 |
| | Male | 926 | _18-24_ | 427 (46.11) | **1.00 (Reference)** | -------- |
| | | | _24-34_ | 238 (25.70) | 2.58 (1.43-4.65) | 0.002 |
| | | | _34-44_ | 118 (12.74) | 6.89 (3.78-12.56) | <0.001 |
| | | | _44-54_ | 58 (6.26) | 6.15 (2.92-12.95) | <0.001 |
| | | | _>55_ | 85 (9.18) | 13.53 (7.31-25.04) | <0.001 |
| _HIV Positive_ | Female | _247_ | _18-24_ | 26 (10.53) | **1.00 (Reference)** | --------- |
| | | | _24-34_ | 64 (25.91) | 2.32 (0.70-7.65) | 0.166 |
| | | | _34-44_ | 92 (37.25) | 3.54 (1.13-11.10) | 0.031 |
| | | | _44-54_ | 46 (18.62) | 3.87 (1.15-13.06) | 0.029 |
| | | | _>55_ | 19 (7.69) | 6.11 (1.51-24.66) | 0.011 |
| | Male | _79_ | _18-24_ | 2 (2.47) | **1.00 (Reference)** | --------- |
| | | | _24-34_ | 8 (9.88) | 0.32 (0.04-1.30) | 0.095 |
| | | | _34-44_ | 21 (25.93) | 0.40 (0.12-1.33) | 0.135 |
| | | | _44-54_ | 27 (33.33) | 0.51 (0.17-1.59) | 0.249 |
| | | | _>55_ | 23 (28.40) | _Omitted due to collinearity_ | -------- |

between men and women (Fig 1). For all GEMS items except for Items 3, 9 and 10 (S2 Table), the majority of men and women reported strong agreement with the statement. Internal consistency does vary with adaption of the GEMS tool, however remained very high in the context of this study (Cronbach's alpha=0.88).

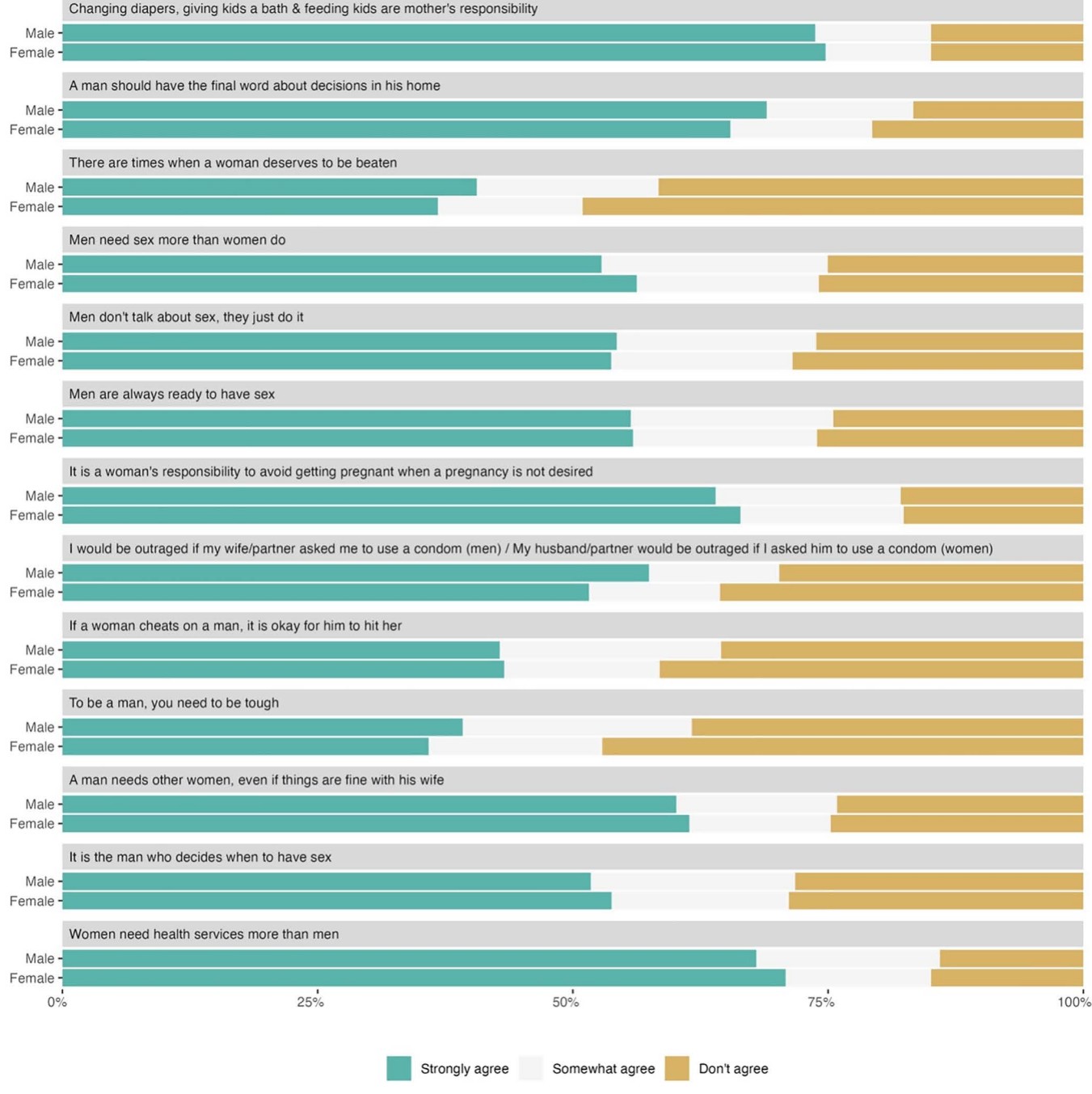

**Fig 1. Responses to each item on the adapted Gender Equitable Men Scale, by sex.**

## Exploratory and confirmatory factor analysis

Initial factoring of the 13 GEMS items revealed high uniqueness (>0.70) values for Item 1, Item 8 and Item 13 (diaper changing responsibility, condom usage, and women needing healthcare more, respectively). Cronbach's alpha of the retained 10 items remained high at 0.87. Factor output from the polychoric correlation matrix using principal factor extraction with the remaining 10 variables demonstrated one factor with an eigenvalue >1. Examining the scree plot (S2 Fig), it was evident the point of inflection ("elbow") of the curve is located at factor 2, where there is a steep rise in eigenvalue magnitude at factor 1. Factors one and two confidently exceed the threshold defined by the parallel analysis (S2 Fig). The resulting rotated factor loadings are given in S3 Table.

Questionnaire items loaded on to factor 1 appeared to relate to sexual autonomy and decisions. Conversely, factor 2 encompassed violence and physical toughness. They are termed "sexual factors" and "violence factors" in the analyses to follow. The internal consistency of both subscales was high, with alpha values of 0.84 and 0.72, respectively. The standardized SEM indicated good fit (RMSEA <0.05, CFI and TLI > 0.90; S2 Table).

## Univariate analysis

All variables were assessed for their potential association with GEMS composite scores and self-reported TB testing status. There was minimal difference in the odds of reporting having ever tested for TB between males and females (0.195 vs 0.184, respectively) with males having a 6% relative increase in odds compared to females, although these findings were not significant (OR = 1.06, 95% CI: 0.86-1.31, p = 0.581). For age group, stratum-specific odds ratios for the outcome clearly demonstrated increasing odds of self-reporting ever testing for TB with increase in age group, with the oldest age group (55 + years of age) having 8.20 times the odds of reporting testing for TB compared to the 18–24-year-old age group (95% CI: 5.53-12.14, p < 0.001). P-values provided strong evidence against the null hypothesis of no difference in odds (<0.001) for all age groups.

Those who were identified as positive for HIV had odds of reporting ever testing for TB nearly three-fold (OR: 2.82, 95% CI: 1.88-4.24, p < 0.001) that of those who were defined as HIV negative. These results are supported by strong evidence against the null hypothesis of no difference in odds of reporting ever testing for TB between HIV groups.

## Multivariate analysis

In the logistic model that included GEMS score, TB testing history, and all covariates, we did not identify a statistically significant association between participants' GEMS score and their likelihood of having previously tested for TB (OR = 1.11, 95% CI: 0.87-1.43, p = 0.396) (see S5 Table). Increasing age group was strongly associated with prior TB testing (p < 0.001), as was HIV status (p < 0.001). Furthermore, a likelihood ratio test (LRT) for age provided evidence of a departure of linear trend (p = 0.004), implying a potential non-linear relationship between age and previous TB testing history. A stratified analysis by HIV status demonstrated people living with HIV (PLHIV) had 50% higher odds of having previously tested for TB with each increasing unit of GEMS score, however this association was not statistically significant (p = 0.088) (Table 2). Inclusion of HIV status as a covariate in the model did not have a significant effect on the association between GEMS score and previous TB testing. The addition of HIV status decreases the odds ratio of previous TB testing from 1.17 (p = 0.125) to only 1.12 (p = 0.262).

The probability of previous TB testing among women increased with age (Fig 2). The probability of previous TB testing did not substantially vary by GEMS score in any age group for women.

In men, the probability of reporting TB testing followed a similar distribution to women with respect to age. The probability of testing increased steadily across all age groups. However, there was more variability regarding GEMS score within age groups for men. In younger men (18 years of age), GEMS score had little effect on the probability of testing for TB. Among men 25 years old, the probability of testing for TB increased with GEMS score (less equitable gender attitudes).

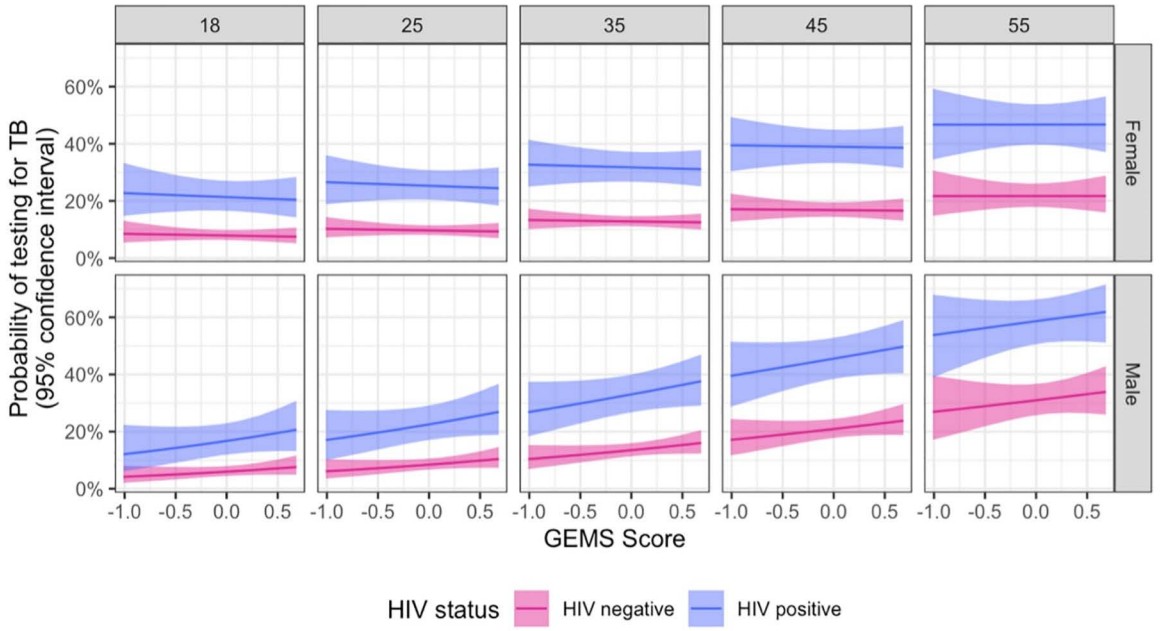

**Fig 2. Predicted probability (with 95% confidence intervals) of ever testing for TB, by GEMS score, age, HIV status, and sex.**

This pattern of increase was seen across all subsequent older age groups of men. This trend is illustrated in Fig 2 through predicted probabilities at indicative ages for men and women, and a summary of the model fit is presented in S5 Table. HIV-disaggregated trends are also shown in Fig 2.

## Discussion

Our analysis of attitudes toward equitable gender norms and self-reported TB testing behaviour found that negative attitudes towards gender equitable norms were prevalent in both men and women in Malawi. Overall, we did not detect a statistically significant association between attitudes towards gender equity, as measured by GEMS, and the odds of self-reporting ever testing for TB. However, among men, we identified a potential effect of age on GEMS score and the probability of previous TB testing. For men, those with higher GEMS scores (less gender-equitable views) were more likely to have tested for TB across age groups. It is notable that no associations between GEMS score and previous TB testing were noted among women.

The potential association between less gender-equitable attitudes and increased likelihood of TB testing history among men could reflect increased risk behaviours that increase the probability of developing TB and, in turn, being tested [3,37–39]. However, it is notable that associations between GEMS score and TB testing history were consistent for HIV-positive and HIV-negative participants across age groups. As behavioural factors were not captured in the extended questionnaire, our interpretation remains speculative and requires further investigation. However, the latent factors extracted from this factor analysis agree with the themes and ideas expressed in existing literature exploring the drivers of sex differences in TB health-seeking behaviours. Sexual factors have been cited in previous qualitative work investigating drivers of the sex differences in health service access [6]. Sexual factors were mentioned in the context of a man's need for sex, and how a diagnosis of TB can impede opportunities for sexual activity in their relationship [6]. Themes of physical violence were less explicitly

stated in literature, however the GEMS items underpinning this latent factor are supported. The idea that toughness is inherent to masculinity is ubiquitous, and TB is understood by individuals to threaten or emasculate men [5,6]. Our findings are broadly consistent with this qualitative literature in underscoring the salience of masculine norms – particularly toughness and control – as barriers to timely health seeking. Our study differed from the prior work in that our quantitative analysis did not identify a clear link between gender attitudes and TB testing. This divergence may highlight how different dimensions of gender norms might operate across study designs. In contrast, among women, we did not observe an association between gender attitudes and TB testing behaviour. We expect that attitudes towards gender equitable norms would have less impact on TB testing among women, given healthcare seeking behaviours are generally more acceptable for women than men [12,40,41]. Whereas gender norms push men towards stoicism and self-reliance [5,6,15], these same pressure are not applied to women, who are accepted as vulnerable and dependent by many hegemonic beliefs.

Our results also suggest an increase in history of previous TB testing with age and HIV status, which is consistent with accumulated testing opportunities over the life course and increased opportunities for TB testing for individuals in HIV care [42–44]. However, age and greater cumulative opportunity for testing may only partially explain our results, as trend assessment signaled variation across age categories that was not strictly linear. This may indicate that factors other than simple time at risk – such as generational differences or historical changes in service accessibility – may also shape testing behaviour across age groups.

To our knowledge, this is the first study to assess the impact of attitudes towards gender equality and gender norms on a TB health outcome. The use of the GEMS tool to assess gender attitudes has been established in the fields of HIV, reproductive health, and intimate partner violence [45–47]. The strengths of the tool in this literature include a validated framework for quantifying attitudes that are often difficult to capture, and evidence that changes in GEMS scores correlate with meaningful improvements in health outcomes and reductions in gender-based violence. However, GEMS has primarily been applied to conditions where gender inequity is a direct determinant of risk or service use, and its adaptation to new disease contexts requires careful consideration of cultural appropriateness and relevance of scale items. Tuberculosis shares some features with conditions for which GEMS has previously been used, particularly related to stigma and gendered barriers to care, yet also differs in that biological and structural risk factors (such as HIV status, poverty, or occupational exposure) may play a large role with TB. The application of GEMS to TB therefore provides an opportunity to explore whether gender attitudes extend their influence on this domain, while also highlighting the need for additional research to determine its validity and utility in the context of TB.

Given the cross-sectional and exploratory nature of our analysis, findings should be considered provisional, and further research is warranted to explore potential links between gender equitable attitudes and TB health-seeking behaviours in men. While no significant associations were found in our study, differences between men and women were notable in our analysis of associations across age groups. Results suggest gender norms may play a role in men's engagement with TB prevention and care, but it is possible that other dimensions of gender that are not assessed by the GEMS tool are more directly relevant. These may include factors more closely linked with documented structural access barriers, perceptions of symptom severity, or the harmonization of routine care with TB care [37–39].

This study was limited by binary structure of self-reported TB testing history, which may distill associations and may be impacted by recall and social desirability bias. It may be worthwhile to evaluate this association in a rural setting, given that this study included only residents within the urban city of Blantyre. Important cultural differences may exist between rural and urban settings that change the nature of gender norms as well as health access behaviours so the generalizability of these findings to rural contexts may be limited. Future work may consider this relationship in global settings, such as Asian and South American regions. Measuring GEMS scores at a single point in time and TB testing retrospectively services as a limitation, as attitudes towards both may have evolved over time. This calls for more data on how gender norms held in these settings may change over time, potentially through repeated surveys or longitudinal study designs relating gender norms to health outcomes; analysis would benefit from further detail on the temporal alignment of individuals'

TB testing with individual and societal shifts in gender norms. Finally, possibilities of accurate exposure ascertainment should be critically evaluated. Further validation work is needed for the GEMS tool against prospectively measured health outcomes such as morbidity, mortality, treatment access, and quality of life. As a complex and subjective social construct, questions surrounding how to best measure gender in this population and setting should be topics of ongoing discussion.

While gender-specific and gender-sensitive services designed to reach men are needed to improve access to diagnosis and treatment [4], our work suggests a link between gender norms and TB testing behaviours among men. Community and policy level interventions may include structural efforts to shift gender equitable norms, potentially increasing willingness to seek TB testing. These interventions would target the upstream determinants of TB risks such as social norms, education, and health system design to improve gender-equitable norms and improve willingness to test for TB. Gender transformative interventions intended to improve health behaviours have previously demonstrated success in reducing intimate partner violence and HIV risk in African settings [45–47] and may have far reaching effects for engagement with TB health services. While more nuanced research into gender norms in the context of TB is needed, our current understandings of the normative constraints that men experience around vulnerability, illness, and help-seeking show a need for gender-responsiveness in TB prevention and care.

## Supporting information

**S1 Fig. Flow Diagram of Study Population.**
(TIFF)

**S2 Fig. Scree Plot of Eigenvalues with Parallel Analysis\*.**
(TIFF)

**S1 Table. Characteristics of participants by HIV status.**
(DOCX)

**S2 Table. GEMS Scale Questions and Responses, by Sex.**
(DOCX)

**S3 Table. Rotated Factor Loadings and CFA Goodness of Fit Statistics\*.**
(DOCX)

**S4 Table. Exploratory Obliquely Rotated Factor Loadings and Unique Variances (items 1, 8 and 13 excluded).**
(DOCX)

**S5 Table. Multivariable Regression Analysis Final Output.**
(DOCX)

**S6 Table. Interaction Model Coefficients.**
(DOCX)

## Author contributions

**Conceptualization:** Elizabeth Di Giacomo, Emily S. Nightingale, Peter MacPherson, Helena R.A. Feasey, Katherine C. Horton.

**Data curation:** Helena R.A. Feasey, Rebecca N. Soko, Vincent K. Phiri.

**Formal analysis:** Elizabeth Di Giacomo.

**Funding acquisition:** Elizabeth L. Corbett.

**Investigation:** Elizabeth Di Giacomo, Peter MacPherson, Helena R.A. Feasey.

**Methodology:** Elizabeth Di Giacomo, Emily S. Nightingale, Peter MacPherson, Helena R.A. Feasey, Katherine C. Horton.

**Supervision:** Emily S. Nightingale, Peter MacPherson, Elizabeth L. Corbett, Katherine C. Horton.

**Validation:** Emily S. Nightingale, Peter MacPherson, Katherine C. Horton.

**Visualization:** Peter MacPherson.

**Writing – original draft:** Elizabeth Di Giacomo.

**Writing – review & editing:** Elizabeth Di Giacomo, Emily S. Nightingale, Peter MacPherson, Elizabeth L. Corbett, Katherine C. Horton.

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
