## [Decision Letter · Decision Letter 0]

23 May 2025

PGPH-D-25-00966

Perceived Gender Equitable Norms and Previous Tuberculosis Testing in Malawi: A Secondary Analysis of a Cluster-based Prevalence Survey

Dear Elizabeth,

Thank you for submitting your manuscript to PLOS Global Public Health. After careful consideration, we feel that it has merit but does not fully meet PLOS Global Public Health’s publication criteria as it currently stands. Therefore, we invite you to submit a revised version of the manuscript that addresses the points raised during the review process.

We look forward to receiving your revised manuscript.

Kind regards,

Collins Otieno Asweto, PhD

Academic Editor

Journal Requirements:

Reviewers' comments:

Reviewer's Responses to Questions

**Comments to the Author**

1. Does this manuscript meet PLOS Global Public Health’s publication criteria?

Reviewer #1: Yes

Reviewer #2: Yes

2. Has the statistical analysis been performed appropriately and rigorously?

Reviewer #1: Yes

Reviewer #2: Yes

3. Have the authors made all data underlying the findings in their manuscript fully available (please refer to the Data Availability Statement at the start of the manuscript PDF file)?

Reviewer #1: Yes

Reviewer #2: Yes

4. Is the manuscript presented in an intelligible fashion and written in standard English?

Reviewer #1: Yes

Reviewer #2: Yes

Reviewer #1: Comments

This manuscript presents a timely and important secondary analysis of a community-based TB prevalence survey in Blantyre, Malawi. The study investigates the association between perceived gender norms—as measured by a modified version of the Gender Equitable Men Scale (GEMS)—and previous TB testing history. The topic is highly relevant given the global push for gender-transformative approaches to TB care, particularly in settings where men remain under-diagnosed and under-treated. The paper is well-written, methodologically sound, and clearly contributes to the growing literature on social determinants of TB health-seeking behavior. However, there are several areas that require clarification and strengthening to enhance the manuscript’s rigor, clarity, and contribution to the field.

1. The rationale for hypothesizing that more equitable gender attitudes would correlate with higher TB testing, particularly among men, is compelling but somewhat underdeveloped. Consider expanding the theoretical framework to include evidence from similar health behaviors (e.g., HIV testing) and gender-transformative health models in Sub-Saharan Africa.

2. The direction of association—where less gender-equitable views among men are linked to higher TB testing—deserves deeper interpretation. The current discussion hypothesizes that this may reflect riskier behaviors (e.g., smoking, alcohol use), but this claim is not empirically explored in the dataset. If data on behavioral risk factors exist in the parent study, consider incorporating them. Otherwise, clearly flag this as a hypothesis and discuss its implications cautiously.

3. The finding that GEMS scores were not associated with TB testing among women, while they were among men, is interesting and should be emphasized further in the discussion. How might gender norms affect men and women differently in health-seeking? This distinction may also have implications for designing gender-transformative interventions.

4. The limitations section is thorough, but could be strengthened by reflecting on the challenge of measuring dynamic social norms (like gender attitudes) with cross-sectional tools. Readers would benefit from a more explicit reflection on how temporality and context may shape GEMS responses and TB testing behavior differently.

5. The authors mention using both exploratory and confirmatory factor analysis for the GEMS tool. Please clarify whether the same dataset was used for both, and whether split-sample validation was considered. Also, indicate the rationale for item exclusion thresholds (uniqueness > 0.70).

Reviewer #2: The manuscript is a robust secondary analysis that explores the association between gender equitable norms and TB testing history within an urban Malawian setting. A modified version of the Gender Equitable Men Scale (GEMS) used within the framework of a population-based prevalence survey adds complexity and richness to the literature in the field of gender and infectious disease epidemiology. The manuscript is methodologically strong, presenting high-level exploratory and confirmatory factor analyses and logistic regression modeling. The authors have stratified by sex and age and have examined three-way interactions, which indicates a thorough and considered analytical strategy. The interpretation of both significant and non-significant outcomes is well-supported and balanced.

Notably, the work adds to the insight into the impact of attitudes toward gender norms on TB service use, particularly among men—a traditionally unaddressed population within TB programming. Although the overall population's primary association between GEMS scores and TB testing was nonsignificant, the trend between TB testing and increasing age among men presents a compelling rationale for designing targeted public health initiatives. The discussion further rests on existing literature and identifies significant applications to gender-transformative TB case-finding strategy design. The conduct and submission of the study are also consistent with journal data transparency and ethics policies, and data are made available through the Open Science Framework and after gaining proper ethical approvals.

Several minor recommendations are made to enhance the manuscript. Of first importance, it would be helpful to specify whether the GEMS score items received equal weights when creating the composite variable, particularly when considering the factor structure. Second, the discussion would be facilitated by a more obvious recognition of the urban context for the study and what it means for generalizability. Third, whereas the authors reference cultural factors for thinking about masculinity and healthcare access, they may want to elaborate further on the ways that such insights might be used to shape community or policy-level intervention beyond the level of individual behavior. These suggestions would further enrich what is already essential and clear writing.

**Do you want your identity to be public for this peer review?** For information about this choice, including consent withdrawal, please see our Privacy Policy

Reviewer #1: No

Reviewer #2: **Yes: ** Abimbola Adegoke

---

## [Decision Letter · Decision Letter 1]

30 Jul 2025

PGPH-D-25-00966R1

Perceived Gender Equitable Norms and Previous Tuberculosis Testing in Malawi: A Secondary Analysis of a Cluster-based Prevalence Survey

Dear Elizabeth,

Thank you for submitting your manuscript to PLOS Global Public Health. After careful consideration, we feel that it has merit but does not fully meet PLOS Global Public Health’s publication criteria as it currently stands. Therefore, we invite you to submit a revised version of the manuscript that addresses the points raised during the review process.

We look forward to receiving your revised manuscript.

Kind regards,

Collins Otieno Asweto, PhD

Academic Editor

Journal Requirements:

Additional Editor Comments (if provided):

Reviewer's Responses to Questions

**Comments to the Author**

Reviewer #3: All comments have been addressed

Reviewer #4: (No Response)

publication criteria?

Reviewer #3: Yes

Reviewer #4: Yes

3. Has the statistical analysis been performed appropriately and rigorously?

Reviewer #3: Yes

Reviewer #4: Yes

4. Have the authors made all data underlying the findings in their manuscript fully available (please refer to the Data Availability Statement at the start of the manuscript PDF file)?

Reviewer #3: Yes

Reviewer #4: Yes

5. Is the manuscript presented in an intelligible fashion and written in standard English?

Reviewer #3: Yes

Reviewer #4: Yes

Reviewer #3: This is a promising and well-designed study, but the manuscript would benefit from deeper theoretical engagement with the findings, clearer discussion of null results, and a stronger articulation of how this work contributes to the broader literature on gender and TB. Enhancing these aspects will improve the manuscript’s impact and clarity.

Reviewer #4: This study is well designed and executed. The topic and the findings are relevant to the field of TB prevention and control and global health more broadly.

The discussion does not adequately address two key issues, however. The association noted between TB testing and age is not surprising since the variable is measured as "ever tested." The authors note that the association is not linear, but provide not further commentary on variation in testing status by age category. If they are want to discuss this finding, they should include some commentary on whether the data suggest that any more complex relationship between age and "ever tested" exists than the simple passage of time.

More significantly, the discussion should reflect more on the practical implications of the lack of an association between GEMS score and testing status. The conclusions state merely that more nuanced research is needed. If, in fact, there is no association between gender equitable attitudes and health service seeking behavior, this would have implications for program design. These findings may indicate that the topic merits more research. At the same time it may also mean that, in fact, gender norms do not have the hypothesized effect on health service seeking behavior and should not be considered a lever for that particular change. The discussion should reflect a consideration of this possibility.

**Do you want your identity to be public for this peer review?** For information about this choice, including consent withdrawal, please see our Privacy Policy

Reviewer #3: **Yes: ** Prashant Subhash Kulkarni

Reviewer #4: **Yes: ** Dora Ward Curry

---

## [Decision Letter · Decision Letter 2]

11 Nov 2025

PGPH-D-25-00966R2

Perceived Gender Equitable Norms and Previous Tuberculosis Testing in Malawi: A Secondary Analysis of a Cluster-based Prevalence Survey

Dear Dr. Di Giacomo,

Thank you for submitting your manuscript to PLOS Global Public Health. After careful consideration, we feel that it has merit but does not fully meet PLOS Global Public Health’s publication criteria as it currently stands. Therefore, we invite you to submit a revised version of the manuscript that addresses the points raised during the review process.

We look forward to receiving your revised manuscript.

Kind regards,

Helen Howard

Staff Editor

Journal Requirements:

1. Please provide a detailed online Financial Disclosure statement. This is published with the article. It must therefore be completed in full sentences and contain the exact wording you wish to be published.

a) State the initials, alongside each funding source, of each author to receive each grant. For example: “This work was supported by the National Institutes of Health (####### to AM; ###### to CJ) and the National Science Foundation (###### to AM).”

For more information, please go to our submission guidelines:

https://journals.plos.org/globalpublichealth/s/submission-guidelines#loc-financial-disclosure-statement

2. Please ensure that the funders and grant numbers match between the Financial Disclosure field and the Funding Information tab in your submission form. Note that the funders must be provided in the same order in both places as well.

3. Please upload your main article file as a .doc, .docx or .rtf file.

4. Please provide separate figure files in .tif or .eps format only and remove any figures embedded in your manuscript file. Please also ensure that all files are under our size limit of 10MB. Please leave the figure captions in the manuscript.

5. We notice that your supplementary figures and tables are included in the manuscript file. Please remove them and upload them with the file type 'Supporting Information'. Please ensure that each Supporting Information file has a legend listed in the manuscript before or after the references list.

Additional Editor Comments (if provided):

Reviewers' comments:

Reviewer's Responses to Questions

**Comments to the Author**

Reviewer #3: All comments have been addressed

publication criteria?

Reviewer #3: Yes

3. Has the statistical analysis been performed appropriately and rigorously?

Reviewer #3: Yes

4. Have the authors made all data underlying the findings in their manuscript fully available (please refer to the Data Availability Statement at the start of the manuscript PDF file)?

Reviewer #3: Yes

5. Is the manuscript presented in an intelligible fashion and written in standard English?

Reviewer #3: Yes

Reviewer #3: There are minor issues to be addressed:-

Phrases like "our work suggests" & "demonstrates" need to be consistent

Ensure captions are desfriptive (e.g. figure 2)

References: SOme recent Gender-TB papers could be added to sterngthen positioning.

**Do you want your identity to be public for this peer review?** For information about this choice, including consent withdrawal, please see our Privacy Policy

Reviewer #3: **Yes: ** Prashant Kulkarni

---

## [Editor Report · Decision Letter 3]

16 Dec 2025

Perceived Gender Equitable Norms and Previous Tuberculosis Testing in Malawi: A Secondary Analysis of a Cluster-based Prevalence Survey

PGPH-D-25-00966R3

Dear Ms. Di Giacomo,

We are pleased to inform you that your manuscript 'Perceived Gender Equitable Norms and Previous Tuberculosis Testing in Malawi: A Secondary Analysis of a Cluster-based Prevalence Survey' has been provisionally accepted for publication in PLOS Global Public Health.

Best regards,

Julia Robinson

Executive Editor